# Phototaxis Characteristics of *Lymantria xylina* (Lepidoptera: Erebidae)

**DOI:** 10.3390/insects16040338

**Published:** 2025-03-24

**Authors:** Jifeng Zhang, Baode Wang, Rong Wang, Xiancheng Peng, Junnan Li, Changchun Xu, Yonghong Cui, Mengxia Liu, Feiping Zhang

**Affiliations:** 1Forestry College, Fujian Agriculture and Forestry University, Fuzhou 350002, China; zhangjif118@163.com (J.Z.);; 2US Department of Agriculture, Animal and Plant Health Inspection Service, S&T, Forest Pest Methods Laboratory, Buzzards Bay, MA 02542, USA; 3Fujian Academy of Forestry Sciences, Fuzhou 350012, China; 4Zhejiang LOIHOI Agriculture and Forestry Technology Co., Ltd., Taizhou 318058, China

**Keywords:** *Lymantria xylina*, light-trapping effect, phototaxis, light-trapping rhythm, sexual dimorphism, fecundity, dispersal risk

## Abstract

*Lymantria xylina* Swinhoe (Lepidoptera: Erebidae), a significant forestry pest in southeastern coastal areas of China, is a regulated invasive insect in some countries. Evidence suggests that *L. xylina* females exhibit phototaxis and possess certain flight abilities, raising concerns about their potential to be attracted to and fly to vessels docked at ports, and egg masses or adults have been intercepted on freighters arriving at a few seaports in the United States and Korea. However, the specific phototactic characteristics and sensitive wavelengths of *L. xylina* remain unclear. To address this knowledge gap, we conducted a comprehensive study on the phototaxis of *L. xylina* adults. We employed different wavelengths and powers of insecticidal lamps to trap adults in the field and tested their phototactic response in various reproductive states in a darkroom. By combining these field and laboratory experiments with data on the weight and egg-carrying capacity of light-trapped female moths, we aim to elucidate the phototaxis characteristics and dispersal ability of *L. xylina*. This research will add more information for evaluating the risk of *L. xylina* reaching vessels through phototaxis flight and spreading long distances via ocean-going vessels.

## 1. Introduction

Phototaxis, a fundamental behavioral response to light stimuli by most insects, plays a pivotal role in insect behavior, influencing navigation, host and food finding, dispersal, mating, and predator avoidance [1,2,3]. Positive phototaxis is exploited in light trapping, a key tool in Integrated Pest Management (IPM) and has become an important monitoring and pest control tool. By attracting and capturing insects, light traps offer several advantages, including delaying pesticide resistance, reducing control costs, and minimizing environmental impacts [4,5].

*Lymantria xylina* Swinhoe, commonly known as the casuarina moth, is a lepidopteran pest of the family Erebidae. It is also known as the casuarina tussock moth, xylina tussock moth, acacia tussock moth, and dark-margined tussock moth [6,7,8,9]. Adult *L. xylina* exhibits typical sexual dimorphism, with apparent differences in the morphology, body weight, and potential flight ability between male and female moths [10,11,12]. It is primarily distributed in Japan (Kyushu and Okinawa, Japan), India, and regions of China, such as Fujian, Taiwan, Guangxi, Guangdong, and Zhejiang. Its extensive host plants pose significant potential risks, as it can damage 424 plant species and shrubs across 103 families (3 subfamilies), including 58 economic crops. In regions like Fujian and Taiwan, *L. xylina* is one of the primary pests of *Casuarina equisetifolia* Forst and *Acacia confusa* Merr; it hinders the normal growth of trees, accelerates the weakness of trees, and even causes the death of whole trees [10,13]. In Fujian, China, *L. xylina* adults generally begin to emerge at the end of May, show peak emergence in early June, and end in late June [14]. In Fuzhou in 2017, the emergence period lasted about 35 days, with the initial period from 23 May to 30 May, the peak period occurring from approximately 8 June to 12 June, and the final period from 19 June to 27 June [12].

Due to its taxonomic relationship with the Asian subspecies of *Lymantria dispar* Linnaeus and similar biological characteristics, *L. xylina* is considered a major invasive pest, gaining international attention and being considered to pose a potential risk of biological invasion [15,16]. Both male and female *L. xylina* exhibit phototactic behavior, with males capable of long flight and some females capable of shorter flight. Females may be attracted to ships’ lights and subsequently land on ships or cargo to lay eggs, resulting in long-distance movement to a suitable habitat with the cargo or ship [12,17]. Studies have shown that *L. xylina* may have a severe invasion risk to subtropical areas in the United States, such as Hawaii, Florida, and Southern California [18,19]. While female *L. dispar dispar* is typically flightless, *L. xylina* females have the potential for limited flight, suggesting a greater dispersal capacity than that of *L. dispar dispar* [10,20,21,22]. Since 2014, several interceptions of *L. xylina* adults and egg masses have been found on ships in some ports in Korea, and egg masses have also been found on vessels called at the U.S. ports [12,23,24,25], further highlighting the risk of long-distance dispersal via maritime transport. However, the extent of phototactic behavior in female *L. xylina* remains unclear, particularly for the population in mainland China, where no reports of light trapping or phototactic flight behavior in females exist.

The movement of *L. xylina* egg masses via sea transportation poses a significant risk for long-distance dispersal. Female moths, attracted to light sources, may lay eggs on ships or cargo at ports or containers stored at ports [10,17]. While the phototactic behavior of *L. xylina* adults, especially female moths, is crucial to understanding this risk, limited research exists. Studies in *Lychee* orchards in Taiwan have shown that high-pressure mercury lamps can effectively trap both *L. xylina* male and female moths with peak activity occurring between 19:00 and 19:30 [26]. In the coastal areas of mainland China, a large number of *L. xylina* adults can also be attracted using frequency–vibration insecticidal lamps [27]. However, the specific wavelength preferences of *L. xylina* and the complete light-trapping rhythm remain unknown.

Insect compound eyes typically have three types of photoreceptors, with the most sensitive in the ultraviolet A region (315–400 nm), being generally concentrated around 365 nm [3,28]. Based on previous studies on the field light trapping of *L*. *dispar asiatica*, females are more sensitive to wavelengths of 365 nm and 380 nm [29]. To investigate the phototaxis of *L. xylina* and its potential for long-distance dispersal, this study examines the response of moths to insecticidal lamps with five different peak wavelengths—363 nm, 365 nm, 368 nm, 378 nm, and 439 nm—in a *Casuarina* plantation. The phototactic behavior of *L. xylina* adults with different reproductive states was tested indoors. By understanding the factors influencing female moth flight and attraction to light, we can better assess the risk of long-distance dispersal of *L. xylina* via ocean-going freighters and develop effective risk mitigation strategies.

## 2. Materials and Methods

### 2.1. Experimental Site

The main experimental site (Appendix A) was located in Pingtan County, Fuzhou City, Fujian Province, China (119°42′48.05″ E, 25°31′51.71″ N), with an elevation of 7.2 m. It is characterized by a subtropical monsoon climate and flat terrain, making it one of the three major wind channels in Pingtan, where wind speeds can reach 14.6 m/s (level 7) at night. The land has undergone sand-fixation afforestation and is predominantly composed of a monoculture of artificially planted *Casuarina* trees, covering an area of 1333.3 hm^2^. The average tree height and diameter at breast height (DBH) of *Casuarina* trees at this location were 7.93 m and 8.67 cm, respectively [30].

### 2.2. Experimental Materials

Since fluorescent lamps were one of the primary light sources for insecticidal lamps and ship lighting, the light source for insecticidal lamps was chosen to be fluorescent lamps, provided by Shenzhen Guanhongya Optoelectronic Technology Co., Ltd., Shenzhen, China, and Zhejiang LOIHOI Agricultural and Forestry Technology Co., Ltd., Taizhou, China. The wavelengths of the light sources were identified by the Physics Laboratory of the Physics Experimental Teaching Demonstration Center of Fujian Agriculture and Forestry University, Fuzhou, Fujian, China (Table 1).

The electric insecticidal parts were manufactured by LOIHOI (Appendix A). The direct current power supply was a 12 V, 20 Ah battery (6-DZF-20, Tianneng Battery Group Co., Ltd., Huzhou, China), and an alternating current was provided by a battery and inverter (NB500, Beijing Newmine Technology Company, Beijing, China). These electric insecticidal parts were equipped with a 4000 V circular high-voltage grid to electrocute the attracted pests, causing them to fall into a collection box, thus controlling and killing them.

### 2.3. Experimental Methods

#### 2.3.1. Trapping Effects of Insecticidal Lamps with Different Wavelengths

Adults of *L. xylina* were trapped during the late peak (26–28 June and 3–5 July 2018) and peak (4–6 June 2019) period of emergence [12,14,27]. Five types of insecticidal lamps (L360, L365, L370, L380, and L405) were used, one for each lamp. The insecticidal lamps were randomly placed daily, with a hanging height of 2 m and about 50 m apart. The insecticidal lamps were turned on at 19:00 every night and off at 4:00 the next day. The number of male and female moths trapped by each lamp was recorded hourly (across nine time periods).

#### 2.3.2. Trapping Effects of L360 Insecticidal Lamps with Different Powers

Because the L360 insecticidal lamp had the best trapping effects on male moths in 2018, the L360 insecticidal lamps with different powers (15 W, 25 W, and 35 W) was used for trapping at the peak period (5–7 June) and late peak period of emergence (15–17 June) in 2019. One lamp for each type was used, and the same method was followed as in Section 2.3.1. The daily trapping of each lamp was counted.

#### 2.3.3. Weight and Egg-Carrying Capacity Between Light-Trapped and Indoor-Emerged Adults

In 2019, a total of 26 female *L. xylina* were trapped using insecticidal lamps in the forest (Appendix A), and 30 trapped male moths were randomly selected. Moreover, these adults were marked as the light-trapped group (outside). A total of 30 male and 30 non-ovipositing female indoor-emerged moths were randomly selected as the indoor-emerged group (inside), and their pupae were randomly collected from the forest, the same as the light-trapped group. After weighing the adults, each female moth was dissected, and the total number of eggs was counted to determine their egg-carrying capacity.

#### 2.3.4. Observation of Phototaxis Behavior Indoor

From May 28 to 31, 2021, the tests were conducted in a dark room measuring 7.2 m × 6 m × 3 m using an L360 insecticidal lamp (25 W) as the light source, which was placed on one side of the room (Appendix A). The non-ovipositing adults emerged from the pupae collected in the field, and the egg-laying female moths with the attached *Casuarina* twigs were collected from the same field. The collection sites were the same as in Section 2.1. Net cages containing *L. xylina* adults were placed 7.0 m from the light source. The light source was turned on from 21:00 until 4:00, and the movement of male and female moths toward the light source was continuously observed. This study was conducted three times: Test 1 involved 44 males and 43 females (1–3 days old, unmated, not laid eggs), Test 2 involved 89 males and 78 females (1–3 days old, mating status unknown, not laid eggs), and Test 3 involved 30 egg-laying females.

### 2.4. Data Analysis

The daily trapping number was normalized using the formula *Z_ij_
*= *y_ij_*/*m_i_* × 100% to minimize the effects of random variations on the test results, where *Z_ij_* represents the adjusted daily trapping number for each type of insecticide lamp, *y_ij_* represents the daily trapping number for each type of insecticide lamp, m_i_ represents the total daily trapping for all insecticide lamps, i represents the date, and j represents the lamp number.

After calculating the values of *Z_ij_*, the variance was analyzed using SPSS 22.0 software (IBM, Armonk, NM, USA). Duncan’s test was used to make multiple comparisons. The independent sample *t*-test was used to compare the two groups. Linear regression was performed using the Levenberg–Marquardt method.

## 3. Results

### 3.1. Trap Catches of Insecticidal Lamps with Different Wavelengths

#### 3.1.1. Comparison of The Trapping Effects of Insecticidal Lamps with Different Wavelengths on Male and Female *L. xylina*

During the late peak period of emergence, all five insecticidal lamps caught male moths of *L. xylina*, while no female moths were caught (Table 2 and Appendix A). During the peak emergence period (4–6 June 2019), a total of 2911 *L. xylina* adults were caught using five types of insecticidal lamps—2902 males and 9 females—resulting in a female-to-male ratio of 1:322 (Table 2 and Appendix A). A substantial number of male moths were caught using all five insecticidal lamps. The L360 lamp exhibited the highest capture rates for male moths, with the maximum total of 1046 and maximum daily catches of 773 moths (Appendix A). In contrast, female moth captures were minimal across all five lamp types. The maximum total number of females captured using any single lamp was only three (L370), and the maximum daily trapping number was two individuals (L370 and L405). The disparity between male and female captures was significant for each lamp type. The maximum total and daily female-to-male ratios of each insecticidal lamp were 1:168 and 1:62 (L405), respectively. These findings suggest that while the *L. xylina* female exhibits some degree of flight phototaxis in the wild, this behavior is considerably weaker than that of male moths.

The adjusted daily capture rates of male moths using different insecticidal lamps are shown in Figure 1a,b. All lamps captured a substantial number of male moths. The L360 insecticidal lamp demonstrated the highest capture rate, followed by the L370 and L365 lamps. During the peak period of emergence, no significant difference was observed in the number of male moths captured using the different lamps (*F* = 2.593, *df* = 4, *p* = 0.101) (Figure 1a). However, during the late peak period of emergence, the number of male moths captured using the different lamps was significantly different (*F* = 5.367, *df* = 4, *p* = 0.007) (Figure 1b). Notably, the L360 lamp accounted for 31.0% and 39.5% of the total male moth captures during the peak and late peak periods, respectively.

#### 3.1.2. Light-Trapping Rhythm of Male and Female *L. xylina*

Due to insufficient female moth captures by individual insecticidal lamp types, data from all five lamp types were combined to assess female light-trapping rhythm. Table 3 presents the capture numbers of male and female *L. xylina* during the different time periods at the peak period of emergence (*F* = 3.156, *df* = 8, *p* = 0.020). The number of female moths captured during the period 19:00–20:00 was the highest, accounting for 44.4% of the total female captures, followed by the 20:00–21:00 period. Subsequently, female captures declined significantly, with no female moths trapped after 23:00. These findings suggest that female moths exhibit peak sensitivity to these lamps shortly after dusk (19:00–20:00).

Significant differences were observed in the number of male moths captured across different time periods (*F* = 3.156, *df* = 8, *p* = 0.020). The highest capture rate for male moths occurred between 23:00 and 0:00, accounting for 29.5% of the total male captures, followed by the 22:00–23:00, 0:00–1:00, and 21:00–22:00. These results show that the male moths exhibit peak sensitivity to these light sources around midnight.

### 3.2. Comparison of Trapping Effects of L360 Insecticidal Lamps with Different Powers

Capture rates were comparable for the L360 lamps with 15 W and 25 W, while the 35 W lamp caught the least (Figure 2a,b). No significant difference in male moth captures was observed among the different power levels for the L360 lamps during the peak period of emergence (*F* = 0.924, *df* = 2, *p* = 0.447) (Figure 2a). However, during the late peak period of emergence, a significant difference (*F* = 19.922, *df* = 2, *p* = 0.002) was found in male moths captures using L360 lamps with different power levels (Figure 2b). Notably, the 25 W lamp captured no female moths, while the 15 W and 35 W lamps captured only one female moth during the peak period of emergence.

### 3.3. Comparison of Weight and Egg-Carrying Capacity Between Light-Trapped and Indoor-Emerged Adults

The average weight of indoor-emerged female moths (1.03 ± 0.06 g) and light-trapped female moths (0.75 ± 0.05 g) was higher than that of male moths (indoor-emerged group: 0.21 ± 0.05 g; light-trapped group: 0.12 ± 0.02 g) (Figure 3a). In the same sex, there was a significant difference in weight between the indoor-emerged group and light-trapped group (female moth: *F* = 2.379, *df* = 54, *p* = 0.0003; male moth: *F* = 9.428, *df* = 58, *p* < 0.001).

Light-trapped females (383.8 ± 32.3 eggs) exhibited significantly lower egg-carrying capacity compared to indoor-emerged females (685.2 ± 42.4 eggs), with the latter being 1.79 times greater (*F* = 5.312, *df* = 54, *p* < 0.001) (Figure 3b).

A strong linear correlation was observed between the weight and egg-carrying capacity in both groups (indoor-emerged females: *r^2^* = 0.941, *p* < 0.001; light-trapped females: *r^2^* = 0.903, *p* < 0.001). However, the linear slopes indicated a lower egg-carrying capacity for light-trapped females compared to indoor-emerged females at equivalent weights (Figure 3c).

In general, compared with the indoor-emerged female moths, the light-trapped female moths were lighter in weight and lower in egg-carrying capacity.

### 3.4. Phototactic Behavior of L. xylina Adults Indoors

In Test 1, 14 out of 44 male moths flew directly toward the lamp, while 13 males exhibited displacement behavior (flying, crawling, and wing flapping) close to the lamp, resulting in a phototactic response rate of 61.4% (Figure 4). Among 43 non-ovipositing female moths, 7 females exhibited displacement behavior close to the lamp, showing a phototactic response rate of 16.3%. In Test 2, 31 out of 89 male moths flew directly toward the lamp, and 17 exhibited displacement behavior close to the lamp, showing a phototactic response rate of 53.9%. Only 1 of the 78 non-ovipositing female moths flew directly toward the lamp, and 9 females exhibited displacement behavior close to the lamp, showing a phototactic response rate of 12.8%. In Test 3, none of the 30 egg-laying female moths showed a phototactic response.

These results suggested that egg-laying females were not sensitive to light sources and did not respond to phototaxis during oviposition. Non-ovipositing female moths exhibited a low phototactic response rate, averaging 14.6%. However, only 0.6% of them (one female) flew directly toward the lamp. Male moths exhibited much a higher phototactic response rate, averaging 57.7%, and 33.3% (45 males) flew directly toward the lamp.

## 4. Discussion

Non-ovipositing female moths exhibit phototactic flight behavior, albeit significantly weaker than that observed in male moths. The field light-trapping test revealed that all five insecticidal lamp types captured a limited number of female *L. xylina* during the peak emergence period (4–6 June 2019). In contrast, all lamps captured much higher numbers of male moths, with the 363 nm lamp exhibiting the highest capture rates. These field observations were further supported by the results of indoor phototactic behavior tests. Egg-laying female moths exhibited no phototactic response, while non-ovipositing females displayed a low average phototactic response rate of 14.6%, with only 0.6% flying directly toward the lamp. In contrast, male moths exhibited a significantly higher phototactic response rate, with an average of 57.7% and 33.3% flying directly toward the lamp. In addition, female *L. xylina* were relatively sensitive to light sources just after dark (19:00–20:00), while male moths were more sensitive to light sources around midnight.

These findings suggest that sexual dimorphism plays a crucial role in the phototactic behavior of *L. xylina*, with males exhibiting stronger responses likely associated with their greater flight activity and mating behavior. The observed differences in phototaxis between male and female adults highlight the diverse preferences and sensitivities of the two sexes to light sources. Even under identical conditions, male and female insects often exhibit distinct phototactic responses [3,31]. For example, most Noctuidae insects trapped by light in the wild are male, and the number of males is far more than that of females, such as *Spodoptera exigua* Hübner [32]

Light source characteristics, particularly wavelength and intensity, significantly influence insect phototaxis. Insects possess visual proteins sensitive to different wavelengths, including UV, blue, and green light [28,33]. For example, *Drosophila melanogaster* is sensitive to 345 nm, 370 nm, 480 nm, and 520 nm [28], while *Narathura japonica* females exhibit greater sensitivity to 460 nm and 560 nm wavelengths compared to males [34]. Light intensity also plays a crucial role. Within a specific range, increasing light intensity generally enhances insect phototactic behavior [3,35]. However, the optimal light intensity varies among species. For example, *Lymantria dispar asiatica* females initiate flight at light levels below 3 lux, while *D. melanogaster* exhibits optimal flight activity at 7 lux [36,37]. These findings provide a foundation for further investigation into the factors influencing the phototactic behavior of *L. xylina* and other insect species. Due to the limited light sources and the small number of female *L. xylina* trapped in this study, it was still necessary to deeply study the sensitive wavelengths and light intensities, especially their interaction.

Oviposition behavior significantly influences the phototactic flight behavior of female *L. xylina*. Previous studies have shown that both *L. xylina* and *L. dispar japonica* females cease active flight after initiating egg laying [10,38]. Indoor tests confirmed that egg-laying females exhibited no phototactic response to light stimuli. These findings suggest that the decline in flight activity after oviposition likely contributes to the low phototactic response and low capture rates of egg-laying females in light traps.

Mating status also influences female flight behavior. Studies have shown that a high proportion of female moths attracted to light sources in the evening are unmated [18,39]. This suggests that mating activity may influence female flight behavior and subsequent light attraction. Mated females may prioritize seeking suitable oviposition sites over engaging in long-distance flights, which could reduce their exposure to light traps.

Flight capability is a critical determinant of an insect’s ability to reach and be captured using light traps. Flight mill tests have revealed that female *L. xylina* exhibit significantly weaker flight potential compared to males. Their flight duration and distance decline rapidly with age, with most females exhibiting minimal flight activity after four days [10].

Interspecific comparisons further emphasize the importance of flight capability in determining light trap capture rates. Flight mill tests have shown that unmated female *L. dispar asiatica* exhibit greater flight distances than unmated female *L. xylina* [10,40]. This suggests that the lower female-to-male ratios observed in *L. xylina* light traps may be partially attributed to the weaker flight potential of female *L. xylina* compared to both males of the same species and females of other species such as *L. dispar asiatica*.

Phototactic behavior may vary among female *L. xylina* populations from different geographic regions. *L. xylina* exhibits significant geographic variation in host plant utilization, body size, sex ratio, and egg mass size [13,41]. For example, the number of eggs per egg mass in Nantou County, Taiwan [41,42], is significantly lower than the egg-carrying capacity of individual females observed in this study and the number of eggs per egg mass reported for Pingtan County, Fujian [11]. A strong positive correlation exists between body weight and egg-carrying capacity in female *L. xylina*. Given this relationship, it can be inferred that female *L. xylina* in Pingtan County may be heavier and larger than those in Nantou County. Flight mill tests further support this hypothesis, demonstrating that female moths with longer total flight distances tend to be lighter in weight, smaller in size, and exhibit lower egg-carrying capacity. These geographic variations in female *L. xylina* morphology and reproductive characteristics may also be influenced by flight–fecundity tradeoffs. Flight–fecundity tradeoffs are well documented in various insects, particularly those exhibiting wing dimorphism [43,44,45]. In *Mythimna separata*, for example, increasing flight capability requires a tradeoff with reproductive output, leading to reduced juvenile hormone levels and inhibited ovarian development. Conversely, the juvenile hormone content increased, the pre-oviposition period shortened, and the flight capability decreased [46,47,48].

In *L. xylina*, limited energy resources may necessitate tradeoffs between energy allocation to flight and reproduction. This may explain why female *L. xylina* with strong flight potential exhibit lower egg-carrying capacity. Furthermore, female *L. xylina* may prioritize reproductive activities over long-distance flight, resulting in reduced phototactic behavior.

This study provides valuable insights into the factors influencing female *L. xylina* phototaxis. However, further research is necessary to fully understand the extent of geographic variation in female phototactic behavior. This research should include (1) controlled experiments to investigate the influence of specific wavelengths and light intensities on female phototaxis, (2) studies to determine the mating status of female moths captured in light traps in different geographic locations, and (3) comparative studies of flight behavior and phototaxis in *L. xylina* populations from different geographic regions. These studies will provide a more comprehensive understanding of the factors influencing female *L. xylina* phototaxis and contribute to the development of more effective monitoring and control strategies for this important pest.

## 5. Conclusions

This study investigated the phototactic behavior of *L. xylina* through field light-trapping experiments and indoor behavioral tests. The results demonstrated that non-ovipositing female moths exhibits phototactic flight behavior, albeit significantly weaker than that observed in male moths. Male moths exhibited strong phototactic responses, with the 363 nm insecticidal lamp demonstrating the highest capture rates.

The phototaxis of *L. xylina* is influenced by multiple factors, including the wavelength and light intensity of light source and the flight capability of both male and female moths. Female moths are additionally affected by other factors, such as geographical population, reproductive status, and behavior characteristics. Egg-laying females exhibits no phototaxis response, while non-ovipositing females displayed limited phototactic behavior. Furthermore, phototactic behavior may vary among *L. xylina* populations from different geographic regions. These findings highlight the importance of considering species-specific characteristics, including sex, reproductive status, and geographic variation, when developing and implementing effective monitoring and control strategies for *L. xylina*.

The limited phototactic flight capability of female moths, particularly those engaged in oviposition, may have significant implications for the long-distance dispersal of this pest. This study provides valuable insights into the factors influencing the phototactic behavior of *L. xylina* and contributes to a better understanding of its dispersal potential and surveillance using different type of traps.

This information will be crucial for developing more effective monitoring and control strategies for this important pest and for enhancing our understanding of insect phototaxis and its ecological significance.

## Figures and Tables

**Figure 1 insects-16-00338-f001:**
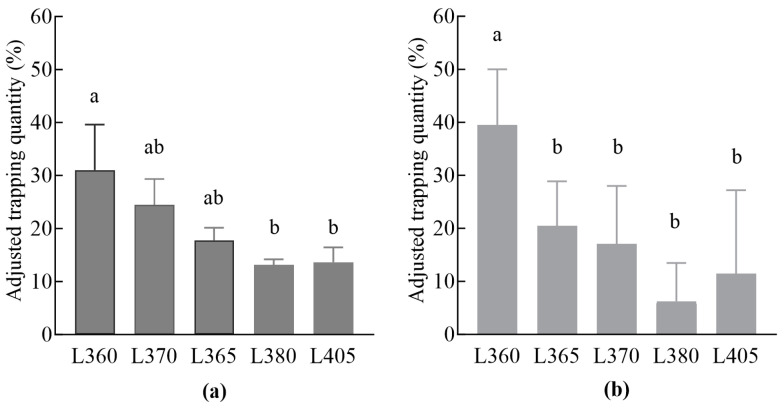
The trapping effects of different insecticidal lamps on *Lymantria xylina* males: (**a**) at the peak period of emergence and (**b**) at the late peak period of emergence. The data in the figure are the mean ± SE; different lowercase letters represent significant differences in the adjusted daily trapping of male moths using different insecticidal lamps (*p* < 0.05, Duncan’s test).

**Figure 2 insects-16-00338-f002:**
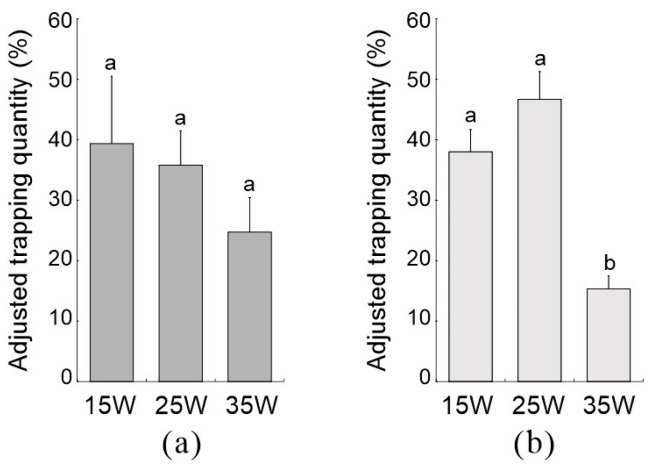
The adjusted daily trapping of L360 insecticidal lamps with different powers on male *L. xylina*: (**a**) at the peak period of emergence and (**b**) at the late peak period of emergence. Data in the figure are expressed as the mean ± SE; different lowercase letters indicate significant differences in adjusted daily trapping of L360 insecticidal lamps with different powers on male *L. xylina* at the level (*p* < 0.05, Duncan’s test).

**Figure 3 insects-16-00338-f003:**
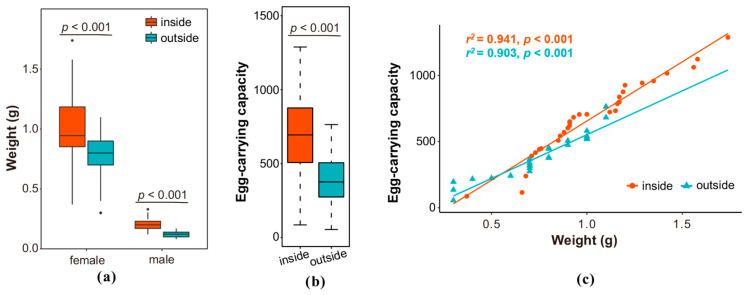
Comparison of weight and egg-carrying capacity between light-trapped and indoor-emerged adults: (**a**) weight, (**b**) egg-carrying capacity, and (**c**) correlation between weight and egg-carrying capacity. An independent samples t-test was conducted to compare the two groups, with black dots indicating outliers. Linear regression analysis was performed using the Levenberg-Marquardt algorithm.

**Figure 4 insects-16-00338-f004:**
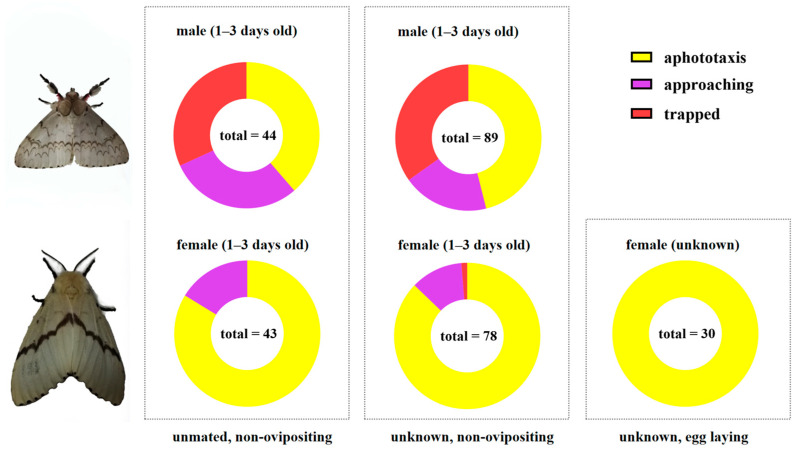
Phototactic response behavior of *L. xylina* adults with different reproductive states indoors.

**Table 1 insects-16-00338-t001:** Configuration and wavelength information of different insecticide lamps.

No.	Central Wavelength (nm)	Voltage (V)	Power (W)	Manufacturer	Electric Insecticidal Part
L360	363	12	15	Guanhongya	GP-LH18B
	363	12	15	LOIHOI	GP-LH18B
	363	220	25	LOIHOI	GP-LH2203
	363	220	35	LOIHOI	GP-LH2203
L365	365	12	15	LOIHOI	GP-LH18B
L370	368	12	15	Guanhongya	GP-LH18B
L380	378	12	15	Guanhongya	GP-LH18B
L405	439	12	15	Guanhongya	GP-LH18B

**Table 2 insects-16-00338-t002:** The number and sex ratio of *L. xylina* trapped using insecticide lamps at the peak and late peak periods of emergence.

Year	Emergence Period	Lamp	Total Trapping			Maximum Daily Trapping		
			Female	Male	Sex Ratio	Female	Male	Sex Ratio
2018	Late peak period	L360	0	16	0	0	4	0
	(6 days)	L365	0	8	0	0	3	0
		L370	0	4	0	0	3	0
		L380	0	4	0	0	2	0
		L405	0	5	0	0	3	0
2019	Peak period	L360	2	1046	1:523	1	773	1:186
	(3 days)	L365	1	502	1:502	1	234	1:212
		L370	3	655	1:218	2	312	1:130
		L380	1	364	1:364	1	200	1:112
		L405	2	335	1:168	2	147	1:62

**Table 3 insects-16-00338-t003:** Comparison of the number of male and female *L. xylina* trapped using insecticide lamps during different periods at the peak period of emergence.

Period	Mean ± S.E.		Cumulative Percent	
	Female	Male	Female	Male
19:00–20:00	1.3 ± 1.3 a	14.3 ± 5.5 c	44.4	1.5
20:00–21:00	1.0 ± 0.6 a	35.0 ± 9.0 bc	77.8	5.1
21:00–22:00	0.3 ± 0.3 a	117.0 ± 37.0 abc	88.9	17.2
22:00–23:00	0.3 ± 0.3 a	212.3 ± 80.9 ab	100	39.1
23:00–0:00	0 a	285.0 ± 89.3 a	100	68.6
0:00–1:00	0 a	179.7 ± 94.6 abc	100	87.2
1:00–2:00	0 a	90.0 ± 54.3 bc	100	96.5
2:00–3:00	0 a	24.3 ± 9.8 c	100	99.0
3:00–4:00	0 a	9.7 ± 4.2 c	100	100

Note: different lowercase letters indicate significant differences in the capture numbers of male and female *L. xylina* during the different periods at the level (*p* < 0.05, Duncan’s test).

## Data Availability

Data are contained within the article.

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
