# Peer review of "Phototaxis Characteristics of *Lymantria xylina* (Lepidoptera: Erebidae)"

_insects, 2025, doi:10.3390/insects16040338_

Round 1

Reviewer 1 Report

Comments and Suggestions for Authors

Manuscript: Phototaxis Characteristics of Lymantria xylina (Lepidoptera Erebidae)

The manuscript provides very interesting results on phototactic behaviour of an insect species, which was investigated both in the field and indoors in controlled conditions.

The work seems to be sound and of value for a scientific target group and will be useful to better control this pest species from spreading.

However, the manuscript needs major revision as it should be improved to reach higher quality, and to increase the reproducibility and scientific rigor of the study.

Introduction

The introduction would benefit from clearly stated research questions and testable hypotheses. This would help the manuscript to follow the outline of these research questions in the methods, results and discussion.

It would be helpful to clarify how the experimental setup relates to lighting used on ships. Currently, outdoor lighting rarely includes wavelengths below 400 nm since fluorescent lamps have been largely phased out.

Materials and methods

The description of the experiments would be strengthened by more clearly describing which lamps were tested on which days and how many replicates were gather from each day.

Critical confounding factors are not reported or included in the statistical analysis but may significantly affect the outcome of the outdoor experiments. I suggest including the weather conditions for each day and hour of the experiment, (temperature, humidity, moon phase and cloudiness, wind speed and direction), potential presence of ambient light sources and amount of light coming from the surrounding in both general and quantitative terms. All these confounding variables may affect the phototaxis of insects.

Technical details of the light sources would be valuable to include (both indoor and outdoor experiments), as these are essential for reproducibility, such as spectral distribution curves (spectral power distribution), light distribution from the lamp and in the surroundings (for example by reporting the ambient luminance or contrasts). Currently, the central wavelength is reported but this does not say much about the spectral distribution of the light source. In fact, two lamps with an identical central wavelength may have very different spectral distribution, resulting in significant different insect attractions. Please clarify whether these light sources had identical spectral power distribution or if they were different. The manufacturer should provide you with this information, or else you can measure it. It would be helpful to explain why these fluorescent lights are used in the experiment if the results should be useful for ships or harbours.

Light exposure levels (lux/intensity or irradiance) for the insects (in their position) should ideally be measured and quantified for the indoor experiments to ensure comparison with other studies. This means reporting the mean value (and the variation) of a metric unit for the whole area insects were contained and where the experiment took place. This would be most useful if measured in directions that is relevant for insect vision, for example, horizontally and vertically against the light source (the maximum values).

Adding photographic documentation of the set-ups, in both outdoor and indoor experiments would help readers more easily understands the methods and visualize the surrounding environmental conditions.

The statistical analysis approach could be enhanced by more properly investigating responses, including interaction effects of intensity and spectra, and confounding variables. A power analysis would give more information about the appropriate sample size. Considering using multivariate analysis or more advanced regression analysis that better can incorporate and handle the experimental design, repeated sampling and confounding factors. The effect sizes and confidence intervals would strengthen the results.

An important consideration is the selection of wavelengths of the lamps included in the study. These are restricted to a very narrow range, between 363-439 nm (“central wavelength”). The range likely falls within measurement uncertainty, especially regarding 363-378 nm. Could you please clarify the justification for using these specific wavelengths choices? What is the rationale for such fine-scale wavelength differentiation? Can insects discriminate between such similar wavelengths? This would be helpful to explained in the introduction. Consider testing more widely separated wavelengths.

The odd effect of intensity/power on phototaxis could potentially be due to confounding factors, and this would benefit from further clarification or better “explanation” by measurement of intensity from different distances from the light source, perhaps the difference in intensity or luminance is not that large from an insect’s perspective.

Another likely possibility worth considering would be that responses to light such as phototaxis may not be a direct linear response to the light source intensity.

Results

Following the lack of research questions, this section is somewhat challenging to follow. It also seems to be excessively descriptive with partly redundant information (in tables, then almost the same in figures). I suggest ensuring that the same kind of information is only reported once.

It would be valuable if the results of insects trapped in the outdoor experiments were also reported per night (number of days) and as total and maximum. Because it is not properly reported how many nights were used for each lamp (and the number of replicates), it is currently not possible to know “daily catch” and compare this study with other studies on insect attraction to light.

Because it is not clear from the manuscript whether the light sources were similar or not in their factual spectral power distribution it is difficult to know if the result for each lamp is relevant to report and test statistically. If the spectra are similar between some of the lamps, you might consider pooling the results, which could lead to a higher number of replicates and more robust results.

The writing structure could be improved, for example avoiding sentences that begins with figure/table references rather than describing biological findings.

The results section would benefit from restructuring after research questions are added and when more proper statistical analysis has been performed.

It is important to include effects of confounding variables.

Discussion

The discussion would be strengthened by centering around the research questions and could be further improved and more focused.

I suggest expanding the discussion of study limitations and alternative explanations for the findings is not considered. For example, consequences of inadequate reporting/knowledge of spectral power distributions of the lamps used in the study and the consequences of not including or even consider confounding factors that may significantly affect the results, in the analysis.

When comparing insect responses to various intensity of the lamps, it would be valuable to report and discuss the light exposure level to be able to compare with previous studies in a scientific manner, or else it is purely speculation.

The discussion could be enriched by referring to other studies showing that there might be sex-differentiated responses to light of insects.

Please avoid referring to figures in the discussion.

Finally, it would be helpful to discuss how these results can be implemented on ships who currently doesn’t use fluorescent light sources.

Reviewer 2 Report

Comments and Suggestions for Authors

This is an interesting study that investigates the phototactic behavior of Lymantria xylina, focusing on the differences between males and females and the implications of phototaxis for long-distance dispersal, particularly via maritime transport. The MS provides a good insight into the species' invasive potential and could contribute to improved monitoring and control strategies. However, I have a few concerns and suggestions for the authors to consider:

  1. The MS shows that male moths exhibit strong phototaxis, whereas female moths demonstrate minimal phototactic behavior. I would like to see a potential explanation for this phenomenon. For instance, could the observed male-biased trapping results be influenced by an already skewed natural sex ratio in xylina populations? If so, the phenomenon is not un-surprising.
  2. The results indicate that reproductive status significantly affects female phototaxis: ovipositing females show no phototactic response (0%), while only 14.6% of non-ovipositing females exhibit phototaxis, and just 0.6% actively fly toward the light. I would like to see further discussion on potential biological mechanisms behind this pattern. For example, could hormonal changes associated with egg-laying suppress phototactic behavior? If there is relevant literature or additional experimental data on this topic, incorporating it into the discussion would strengthen the study’s conclusions and discussion.
  3. Many readers may not be familiar with insect phototaxis behavior. For that, I suggest that the authors provide supplementary materials, such as videos demonstrating the phototactic responses of xylina in controlled experiments.

Round 2

Reviewer 2 Report

Comments and Suggestions for Authors

The reviewer has addressed almost all of my concerns.